# Heterozygote Dopamine Transporter Knockout Rats Display Enhanced Cocaine Locomotion in Adolescent Females

**DOI:** 10.3390/ijms232315414

**Published:** 2022-12-06

**Authors:** Marta Pardo, Michele Martin, Raul R. Gainetdinov, Deborah C Mash, Sari Izenwasser

**Affiliations:** 1Department of Neurology, University of Miami Miller School of Medicine, Miami, FL 33136, USA; 2Department of Psychiatry and Behavioral Sciences, University of Miami Miller School of Medicine, Miami, FL 33136, USA; 3Institute of Translational Biomedicine and St. Petersburg University Hospital, St. Petersburg State University, Universitetskaya Emb. 7-9, 199034 St. Petersburg, Russia

**Keywords:** dopamine transporter, cocaine, knockout, heterozygous, sex differences, addiction

## Abstract

Cocaine is a powerful psychostimulant that is one of the most widely used illicit addictive. The dopamine transporter (DAT) plays a major role in mediating cocaine’s reward effect. Decreases in DAT expression increase rates of drug abuse and vulnerability to comorbid psychiatric disorders. We used the novel DAT transgenic rat model to study the effects of cocaine on locomotor behaviors in adolescent rats, with an emphasis on sex. Female rats showed higher response rates to cocaine at lower acute and chronic doses, highlighting a higher vulnerability and perceived gender effects. In contrast, locomotor responses to an acute high dose of cocaine were more marked and sustained in male DAT heterozygous (HET) adolescents. The results demonstrate the augmented effects of chronic cocaine in HET DAT adolescent female rats. Knockout (KO) DAT led to a level of hyperdopaminergia which caused a marked basal hyperactivity that was unchanged, consistent with a possible ceiling effect. We suggest a role of alpha synuclein (α-syn) and PICK 1 protein expressions to the increased vulnerability in female rats. These proteins showed a lower expression in female HET and KO rats. This study highlights gender differences associated with mutations which affect DAT expression and can increase susceptibility to cocaine abuse in adolescence.

## 1. Introduction

Dysregulations in dopamine (DA) have been shown to play important roles in several diseases, including addiction. DA is an important neurotransmitter that regulates multiple functions including locomotion, emotion and cognition [1]. The DA transporter (DAT), expressed on the surface membrane of DAergic neurons, plays an important role in regulating intracellular and extracellular DA levels to maintain DA homeostasis through DA reuptake. DAT has been widely studied for its role in addiction [2,3,4,5,6]. Major psychostimulants, such as cocaine and amphetamine, increase extracellular DA levels [7,8] by blocking DAT on DAergic nerve terminals and inhibiting DA reuptake (e.g., [9,10]). Due to DATs role in DA regulation, decreases in DAT expression could increase rates of drug abuse and vulnerability to psychiatric disorders due to the associated hyperdopaminergic state, making people with DAT polymorphism susceptible to multiple disorders [2]. 

Previous studies indicate that knockout (KO) of the DAT protein in mice induces functional hyperdopaminergia, aberrant DA biosynthesis and signaling, and an altered response to psychostimulants [9,11,12,13,14,15,16,17]. However, studies with mice have limited translational applicability in human trials [18]. Recently, a new transgenic rat model with a partial (DAT+/− heterozygous, (HET)) or total (DAT −/−, knockout, KO) depletion of the DAT gene has been developed [19,20]. Leo et al. [19] demonstrated a consistent hyperdopaminergic state at molecular, neurochemical, pharmacological and behavioral levels in DAT KO rats. However, a more detailed focus on the HET rats as a model of hyperdopaminergia will allow us to further study the role of and mediation of DAT as it relates to cocaine reward during adolescence.

Although drug addiction affects both sexes, addiction develops and progresses differently in females compared with males [21]. Women report a greater subjective high in response to cocaine, even when drug levels and metabolite production are equivalent across the sexes [22,23]. Significant sex differences in both acute responses and adaptations to repeated drug exposures in adolescents and adults have been reported previously [24,25,26,27,28]. Importantly, women present higher comorbidity of drug abuse with other psychiatric disorders [29]. Additionally, adolescence is a time of increased vulnerability to the effects of drugs [30]. The brain enters a critical period during adolescence and its structure as well as its changes in activity have been widely described [31], making adolescence a particularly vulnerable period for substance abuse as a trajectory to cocaine dependence in adults [32,33]. Teenagers vary in their ability to control impulses that contribute to risk-taking associated with low dopamine [34,35]. Additionally, during adolescence, there are developmental changes in the DAergic system, such as in catechol-O-methyltransferase (COMT), tyrosine hydroxylase (TH), and DA, D2 and D1 receptors (please see [36]), which have been conceptualized as underlying adolescent-specific changes in motivational behavior [37]. 

In the present study, we examined DAT KO in adolescence to determine the effects of lower expressions of DAT in acute and chronic cocaine impacts on rodent locomotion. We examined the behavioral phenotype of DAT KO and HET rats in response to different doses of cocaine, as well as the DA presynaptic markers that are modulators of DA, to test the hypothesis that partial or full deletion of DAT leads to differential changes in the basal DA system which mediates altered behavioral responses to cocaine. Based on the previous literature highlighting the importance of the developmental adolescence stage to addiction, we examined DAT in adolescent male and female rats. The results in DAT heterozygote rats demonstrate that female rats are more sensitive to elevated synaptic dopamine during adolescence compared to males. Female adolescent drug use has increased dramatically in the last 30 years. These results in the rodent model further support the idea that the syndrome of female adolescent substance abuse is different from the well-recognized male pattern.

## 2. Results

### 2.1. Experiment 1: Characterization of DAT HET and KO Rat’s Phenotype and Response to Cocaine Administration

**Response to Novelty**: The response to a novel environment was determined by analysis of the first 10 min in the open field (prior to drug administration, intervals 1 and 2, Figure 1). An overall two-way ANOVA (genotype x sex) showed that there was a significant effect from genotype (F (2,228) = 385.7, *p* < 0.001), but no significant effect from sex and no sex–genotype interaction. Post hoc testing revealed that the KO animals react to novelty with increased distance traveled, when compared to both WT and HET rats (*p* < 0.001), but that there were no significant differences between WT and HET (Figure 1). When WT and HET were analyzed without KO, there were no significant differences in novelty across sex or genotype and no significant interactions. As can be seen in Figure 1, the KO rats of both sexes (Figure 1E,F) exhibited much higher levels of activity than either WT (Figure 1A, B) or HET (Figure 1C,D) rats immediately upon being placed in the open field. 

**Habituation**: To determine whether all the groups of animals had similar baselines immediately prior to being injected with the drug or vehicle, an overall three-way ANOVA of genotype (WT, HET, KO) x sex x dose of cocaine (0, 10, 20 mg/kg) was examined based on activity during the last 10 min of the habituation period (intervals 5 and 6, Figure 1). There was a significant effect of genotype (F (2,216) =463.8, *p* < 0.001) but not of sex or dose. Post hoc testing showed that the KO rats continued to exhibit significantly elevated levels of activity compared to WT and HET rats (*p* < 0.001), even after being in the open field for 30 min. Thus, compared to the WT and HET rats, the KO rats do not appear to habituate to the environment during this period, an effect that can be masked due to the basal hyperactive phenotype of these rats (Figure 1). These data confirmed that the KO rats are hyperactive and have a significantly overall greater baseline activity compared to WT or HET rats (*p* ≤ 0.05). Due to the overall hyperactive locomotor activity in KO rats, for visual purposes in the graphical representation of the data, KO rat results are presented across the data in separate graphs. Appendix A shows one illustrative example of each genotype travel pathway on the open field during the first 30 min.

Effects of acute cocaine administration on locomotor activity in an open field: 

We initially wanted to test the hypothesis that adolescent DAT HET and KO rats are vulnerable to drugs of abuse, such as an acute dose of cocaine. Overall, there was a significant effect from genotype (F (2,216) = 373.4, *p* < 0.001) with post hoc tests showing that the KO rats differed significantly from WT and HET rats (Figure 1A–F). Their activity levels were extremely high, and they did not exhibit differences in activity as a function of cocaine dosing. However, there also was a significant effect from the dose (F (2,216) = 6.83, *p* < 0.001) and a significant dose x genotype effect (F(4,216) = 2.95, *p* < 0.05), because cocaine did not increase activity in the KO rats but did so in the WT and HET rats.

As observed during the novelty and habituation phases, the KO rats maintained a very high level of activity that did not differ significantly in response to saline, 10 mg/kg cocaine, or 20 mg/kg cocaine (Figure 1E,F). They maintained approximately a 10-fold increase in activity compared to WT and HET rats throughout the 60-min test period. 

These initial studies confirmed that the KO rats have extremely high rates of activity, in agreement with DAT KO mice data (Efimova et al., 2016), and that acute administration of cocaine at doses which normally produce increases does not alter the activity significantly. 

It does appear that the extremely high activity of the KO rats renders any differences between WT and HET rats insignificant, thus, additional analyses were done on WT and HET activity in response to both acute and chronic cocaine administration. A three-way ANOVA (genotype x sex x dose) showed a significant effect from sex (F(1,158) = 10.85, *p* ≤ 0.001), with females exhibiting a greater response to cocaine than males overall; from genotype (F(1,158) = 12.54, *p* ≤ 0.001), with HET rats ranking higher overall than WT; and from dose (F(2,158) = 84.08, *p* ≤ 0.001), with 10 and 20 mg/kg leading to higher levels of activity than saline. In both males and females, both doses of cocaine produced significant increases in activity over saline (Figure 1A–D). There also was a significant sex x dose interaction (F (2,158) = 6.96, *p* < 0.001) in WT and HET rats. Post hoc tests showed that there were sex differences at the 10 mg/kg dose of cocaine, with cocaine having a greater effect in WT and HET females than in males (*p* < 0.05). There was a significant genotype x dose effect (F (2,158) = 8.15, *p* < 0.001), with a significant effect from genotype in response to a dose of 10 mg/kg cocaine but not to saline or to 20 mg/kg. This was due to the significantly greater effect of 10 mg/kg cocaine on HET females than on WT females (*p* < 0.05) and of 20 mg/kg cocaine on HET males than on WT males (*p* < 0.05). Similarly, 20 mg/kg cocaine produced a significantly greater effect on HET males than females (*p* < 0.05). 

**Activity over 60 min after cocaine acute administration:**Figure 2 shows the cumulative effects of cocaine across genotypes. A three-way ANOVA (sex x genotype X dose) showed significant effects of cocaine dose (F (1,127) = 8.99, *p* < 0.05), as well as of genotype (F(2,127) = 204.4, *p* < 0.001). Both doses of cocaine significantly differed from saline in WT and HET males and females (*p* < 0.05). 20 mg/kg cocaine induced a higher distance traveled in WT males (Figure 2A) as well as HET females (Figure 2B) compared to 10 mg/kg (*p* < 0.05). However, no difference between doses was found in WT males (Figure 2A) or HET females (Figure 2B). Additionally, as shown in Figure 1, KO rats have extremely high locomotor activity (Figure 1E,F; please be aware of different axes compared to panels A,B,C,D). KO rats have significantly increased locomotor activity compared to WT rats, and show no effects from cocaine doses (Appendix A). KO DAT rats present an accumulated increase in distance traveled at a basal level (saline-treated) of an order of magnitude 32–42 (male) and 15–22 (female) fold of that observed in WT and HET rats (total distance (cm) (saline-treated): WT: 2212 (male) 3931 (female), HET: 2967 (male) 5739 (female), and KO: 94,595 (male), 86,484 (female)). 

Based on these results from the acute studies and previous publications (Mandt et al., 2008) a dose of 10 mg/kg cocaine was selected for the chronic experiment. This was due in part to the very large response to 20 mg/kg cocaine in some conditions and the concern that it would not be possible to observe an increase in activity in response to this dose if the animals became sensitized to the effects of cocaine.

**Chronic effects of cocaine on locomotor activity in DAT rats in the open field:** Rats received repeated daily doses of saline or cocaine (10 mg/kg cocaine) in their home cages from PND36 to PND41. Each rat received the same dose across days. On PND42 rats were habituated to the open field followed by cocaine administration, and measurement of locomotor activity was recorded for the acute study. 

**Habituation during chronic condition:** Analysis of the habituation period showed that there was no significant sex x genotype x dose difference in habituation to the chamber overall (Figure 3, time interval 5 and 6; 3 Way ANOVA). However, genotype had a significant effect, (F (2,127) = 274.56, *p* ≤ 0.001); the KO rats remained non-habituated to the test chamber, reflected by the increased activity in the KO rats (Figure 3E,F). There were no sex differences overall in habituation.

**Effects of chronic 10 mg/kg cocaine**: We focused on the effects on locomotor activity of chronic exposure to 10 mg/kg cocaine for seven days to test the hypothesis that rats with a DAT mutation would show increased vulnerability to the effects of cocaine. A three-way ANOVA showed significant effects from dose (F(2,127) = 9.55, *p* < 0.05) and genotype (F(2,127) = 216.47, *p* < 0.001), but not sex. There also was an interaction between sex and dose (F (1,127) = 4.001, *p* < 0.05). 

Chronic 10 mg/kg cocaine dosing increased the locomotor activity of WT (Figure 3A,B) and HET rats (Figure 3C,D). KO rats did not show any effect from the chronic administration of cocaine (Figure 3E,F). Importantly, chronic cocaine administration had a stronger impact on female WT and HET rats compared to male rats (*p* < 0.01). Furthermore, the effects of chronic cocaine administration were higher in HET rats compared to WT rats, in both sexes (*p* < 0.001). 

**Activity over 60 min after chronic administration:** The analysis of WT and HET rats (without KO rats) showed the significant effects of sex (F(1,91) = 22.24, *p* ≤ 0.001), of dose (F(1,91) = 3.813, *p* ≤ 0.001), and of genotype (F(1,91) = 7.98, *p* ≤ 0.01) on habituation using a three-way ANOVA. There also were significant sex x dose interactions (F (1,91) =5.35, *p* ≤ 0.05) and important trends in genotype x dose (F(1,91) = 3.251, *p* ≤ 0.075). Post hoc analyses showed significant increases in locomotor activity after chronic cocaine administration in male and female WT and HET rats (*p* < 0.05, Figure 4A,B). Additionally, chronic cocaine administration induced significant increases in HET compared to WT, males and females (*p* < 0.05). Finally, the effects of chronic cocaine were higher in females than males. KO rats did not show effects of cocaine administration (Appendix A) due to their high basal levels of activity. 

**Comparison of Acute vs. Chronic Cocaine (10 mg/kg) in WT and HET rats:** Since cocaine did not alter locomotor activity levels in KO, we focused our additional comparisons on WT and HET rats. There were significant differences in the response to chronic cocaine overall in WT and HET rats. A within-subject ANOVA comparison of initial (acute condition) vs. later (chronic condition) administration of cocaine showed a significant effect overall (F (1,91) =7.929, *p* < 0.01) and a time (acute vs. chronic) x sex interaction (F(1,91) =7.10, *p* ≤ 0.01). Between-subject analyses showed that there were significant effects on locomotor activity in response to cocaine treatment, of genotype (F (1,91) =11.23, *p* ≤ 0.001), sex (F(1,91) =20.76, *p* ≤ 0.001) and dose (F(1,91) =52.58, *p* ≤ 0.001). There were also significant sex x dose (F (1,91) =5.54, *p* ≤ 0.05) and genotype x dose (F(1,91) =5.17, *p* ≤ 0.05) interactions. Post hoc testing showed that HET and WT rats did not differ in response to saline but did differ overall in response to 10 mg/kg cocaine (*p* ≤ 0.05), with the HET rats having an overall greater effect from cocaine administration (acute and chronic) on their locomotor activity than the WT rats. There also were significant differences in both WT and HET rats in response to 10 mg/kg cocaine vs. saline (*p* ≤ 0.05). 

### 2.2. Experiment 2: Sensitization of Locomotor Activity to Cocaine in DAT Rats in the Open Field

For a better understanding of the vulnerability that a DAT mutation could have on cocaine relapse, we tested the hypothesis that the effects of cocaine, after a short period of washout, could be even enhanced with acute cocaine exposure.

After being tested on the acute and chronic conditions, rats were left undisturbed for five days. On the sixth day (PND48), rats were exposed to an acute dose of cocaine (saline or 10 mg/kg, hereafter referred to as sensitization challenge). Each rat received the same dose during the acute and sensitization challenge. 

Comparison of WT, HET, and KO:

A four-way ANOVA on all three genotypes with testing (acute vs. sensitization) as a within-subjects (repeated) variable and sex, genotype, and dose as between-subject variables revealed significant interactions between test x sex x dose (F(1,79) = 8.42, *p* ≤ 0.005) and test x genotype x dose (F(1,79) = 4.19, *p* ≤ 0.05) (Figure 5 and Figure 6). There also were significant effects on the testing overall (F (1,79) = 15.66, *p* ≤ 0.001), test x sex (F(1,79) = 13.92, *p* ≤ 0.001), test x genotype (F(2,79) = 18.32, *p* ≤ 0.001), and test x dose (F(1,79) = 7.58, *p* ≤ 0.01). Thus, the changes in response to cocaine during the sensitization challenge vs. the acute response differed as a function of sex, genotype and dose (due to the lack of change in the saline response). 

A three-way analysis (sex x genotype x dose) of the sensitization challenge alone (without being compared to the acute effects, Figure 5) in WT, HET and KO rats revealed a significant effect of genotype (F (2,79) = 120.13, *p* ≤ 0.001) with no significant interactions (HET rats had a higher response to cocaine). KO rats, as on acute and chronic exposure to cocaine, did not show effects of cocaine administration in the challenge alone (Appendix A).

Comparison of WT and HET:

A four-way ANOVA on WT vs. HET with test (acute vs. sensitization) as a within-subjects (repeated) variable and sex, genotype and dose as between-subject variables revealed a significant interaction between test x sex x dose (F (1,56) = 11.81, *p* ≤ 0.001). There also were significant effects of test overall (F (1,56) = 21.78, *p* ≤ 0.001), of test x sex (F(1,56) = 5.27, *p* ≤ 0.025), and test x dose (F(1,56) = 18.60, *p* ≤ 0.001). Thus, the changes in response to cocaine during the sensitization challenge vs. the acute response differed as a function of sex and dose (due to the lack of change in the saline response) but not by genotype. 

The analysis of the sensitization challenge alone (without being compared to the acute effects) in WT and HET rats revealed significant effects from sex (F (1,56) = 35.99, *p* ≤ 0.001) and dose (F(1,56) = 15.54, *p* ≤ 0.001) and sex x dose (F(1,56) = 10.93, *p* ≤ 0.002) (Figure 5 and Figure 6). Female WT and HET rats showed a greater response to the cocaine challenge when compared to their respective males. Additionally, HET female rats showed enhanced and prolonged responses to cocaine (compared to WT female rat). Genotype and genotype x dose just missed significance with *p* ≤ 0.055. 

### 2.3. Experiment 3: Dopamine Pathway Protein Alterations in DAT KO Rats

DAT HET and KO rats have been described as a model of hyperdopaminergia with increased levels of DA. However, other relevant mediators of DA synthesis and release have not been previously described in these rats. Different proteins likely play important roles in mediating the behavioral phenotype in response to cocaine treatment, DAT, tyrosine hydroxylase (TH), vesicular monoamine transporter 2 (VMAT2), α-Synuclein (αSyn) and a protein interacting with C-kinase-1 (Pick1)), were analyzed in the striatum of DAT HET and KO rats. 

Protein analysis demonstrated reduced levels of phosphorylated TH (pTH) in the striatum of DAT KO transgenic male rats (One-Way ANOVA F (2,13) = 9.21, *p* < 0.05) and female rats (One-Way ANOVA F(2,13) = 3.71, *p* < 0.05) (Figure 7A and Appendix A). Post hoc analyses showed that there was a significant reduction in pTH in male KO rats (*p* < 0.05) as well as in female KO rats (*p* < 0.05) compared to their respective WT. No significant changes were found in the pTH levels of HET rats. VMAT2 protein expression levels were also studied across genotype and sex (Figure 7A and Appendix A). There were significant differences in the levels of VMAT2 in male rats (one-way ANOVA F (2,12) = 20.05, *p* < 0.05). Posthoc analyses showed significant decreases in HET (*p* < 0.05) and KO (*p* < 0.05) compared to WT levels. Additionally, a one-way ANOVA showed significant differences in VMAT2 levels in female rats (F (2,13) =12.87, *p* < 0.05) with levels significantly reduced in HET and KO rats (*p* < 0.05) compared to WT. Additionally, the α-synuclein was studied across genotypes (Figure 7C and Appendix A). A one-way ANOVA showed that there was a significant effect based on genotype in males (F (2,13) = 4.60, *p* < 0.05) and females (F(2,13) = 6.55, *p* < 0.05). Post hoc analyses showed a significant reduction in α-synuclein in KO males (compared to WT males, *p* < 0.05). Additionally, female HET and KO rats showed reduced levels of α-synuclein in the striatum when compared to WT female levels (*p* < 0.05). Finally, Pick 1 protein levels were studied (Figure 7D and Appendix A). A One-Way ANOVA for genotype yielded significant differences for the males (F (2,13) = 4.94, *p* < 0.05) and females (F(2,13) = 8.08, *p* < 0.05). Separate posthoc analyses showed a significant reduction only in HET male rats compared to WT males (*p* < 0.05), and a significant reduction in HET (*p* < 0.05) and KO (*p* < 0.05) female rats compared to their respective female WT. 

## 3. Discussion

The present study gives support to the main role of DAT on vulnerability to cocaine use and abuse. This study is the first one to establish the DAT HET and KO rats as a model for the study of cocaine addiction. We show that HET DAT rats respond more strongly to an acute and chronic dose of cocaine and that those effects are maintained for a longer period. We highlight important sex differences since a lower acute dose, as well as chronic cocaine exposure, induced an increase in locomotor activity in HET female rats that doubled the effects of the same cocaine condition in HET male rats, as well as compared to WT female rats. Additionally, we give additional support to the fact that WT females have an enhanced response to cocaine, compared to WT males, in a variety of cocaine doses (10, 20 mg/kg) and cocaine exposures (acute, chronic, sensitization). However, male DAT HET rats showed increased sustained responses to the highest dose of cocaine. Finally, full mutation of DAT (KO rats) represents a model of hyperdopaminergia with an impaired development that induces sustained hyperlocomotor activity under basal conditions. Locomotor activity levels are unchanged with cocaine exposure in KO rats, and cocaine loses functionality in full KO rats. With the additional support of reduced protein levels of TH in KO DAT rats, and VMAT in DAT HET and KO rats, we establish the dopaminergic imbalance that this new rat model presents. However, importantly, sex differences in the protein levels of α-syn and PICK1 highlight the role of those proteins on cocaine-mediated locomotor effects. In conclusion, this new DAT KO/HET rat model represents a new useful tool for the study of addictive behavior, overcoming the limitations of mice models. DAT alterations predispose vulnerability to the effects of cocaine, and warrant further studies. 

Vulnerability to drug abuse is determined by both genetic predisposition and environmental influences. After decades of research and an increased understanding of the adolescent brain, the heightened propensity to develop cocaine addiction in this population is still subject of study. The search for novel pharmacological interventions for cocaine addiction is a need. DAT knockout mice have been studied in the past and have helped to understand the behavioral effects and underlying mechanisms regulating cocaine’s and other psychostimulants’ effects. However, due to the important role that DAT plays on cocaine’s effects, especially during a vulnerable age period such as is adolescence, we still need to further our understanding. The present study used DAT KO rats to focus on DAT’s role in mediating the effects of cocaine, with an emphasis on sex-based differences during adolescence. Adolescence has been described as a vulnerable period in humans for developing cocaine addiction [38,39,40], and laboratory studies also have shown that there is increased vulnerability to the effects of drugs after exposure during adolescence [41,42,43]. Due to adolescents’ state of vulnerability, we studied acute and chronic effects of cocaine during adolescence in rats with a full or partial mutation of DAT. Our data confirm and extend previous results obtained with DAT KO mice. KO rats have exceptionally high levels of activity, and differences in DAT expression affect susceptibility to the effects of cocaine [6,13]. However, for the first time, we used transgenic HET and KO DAT rats to study the effects of cocaine, as well as the underlying mechanisms that could mediate the increased vulnerability to its effects. Our results show that: (1) rats with a mutation of the DAT (50% reduced DAT gene expression (HET)) exhibit an enhanced response to cocaine-induced locomotor effects; (2) genotype-dependent effects of cocaine are, in most of the conditions studied, stronger in female compared to male rats and (3) main regulators of DA are altered due to DAT reduction or full ablation.

Prior studies on the effects of cocaine in DAT KO mice have been published [6,9,13,44,45,46,47,48]. Full deletion of DAT in mice induced several adaptative changes across all DA systems including its synthesis, storage, extracellular levels, and receptor expression and functions [8,9]. However, the focus was mostly on DAT KO mice and truly little information has been reported about DAT HET in response to drugs of abuse. This is especially important because even though we recognize the relevant information that full KO animal models have provided, most of the human pathological conditions where DAT has been involved present a polymorphism or mutation that only induces a reduction in DAT, not a full mutation. Based upon the current data, as well as previous publications, we propose that the HET rats present a more valid model for the study of psychopathology such as addiction. 

As previously characterized in DAT KO mice [9], DAT KO adult rats are hyperactive in the open field [19]. Additionally, locomotor activity during adolescence seems to fluctuate in DAT KO rats, showing twice as much activity during pre-adolescence (PND27) compared to adolescence (PND34) [49]. The present study extends the work of previous publications, without exposure to any drug, across adolescence (PND35-PND48). Over the period studied in the current group of experiments, DAT KO male and female rats maintained increased baseline levels of locomotor activity compared to HET and WT rats. However, those increased levels did not fluctuate at different time points across adolescence. KO male and female rats showed acute temporary reduced locomotor activity after ip administration, which could have resulted from the stress generated by the hand restraint needed for the administration. The vulnerability to the restraint and handling for ip administration agrees with recent data showing increased response to stress in these rats [50,51] and relevant alterations in brain areas relevant for impulsivity and behavioral control (prefrontal cortex) during vulnerable developmental stages such as adolescence [52]. 

The previous results in DAT KO mice showed no effects of acute cocaine in locomotor activity [9,48]. In contrast, DAT KO mice were proposed as a model of ADHD due to the calming effects of psychostimulants; cocaine attenuated hyperlocomotion of KO mice [14]. Additionally, repeated exposure to cocaine in a conditioned place paradigm showed that, not after acute 20 mg/kg treatment but after few administration days, KO mice increased locomotor activity [47]. Previous studies in mice concluded that DAT deletion (full KO mice), led to an increased sensitization to cocaine effects [48]. However, our data show that DAT KO male and female rats, during adolescence, did not alter their increased locomotor activity when treated with cocaine. DAT KO rats present hyperlocomotor activity that is not altered across all study conditions. Future studies will expand the results obtained into adulthood. 

Adinolfi et al. [49] showed no differences in activity in adolescent DAT HET rats (PND34) compared to WT rats. Our data replicate and expand previous results in HET male and female rats. Baseline activity in HET rats did not differ from WT on either PND35, 42, or 48. As seen in DAT HET mice [9], DAT HET rats appear normal. However, their response to cocaine is potentiated. We found that a 50% reduction of DAT expression enhanced the locomotor stimulatory effects of cocaine (10 and 20 mg/kg) in HET male and female rats. Acute cocaine treatment induced a greater response in HET male and female rats, and this effect persisted throughout the hour of recording. An increased response to acute cocaine (10 mg/kg) was previously shown in adult HET mice [48]. However, responses to higher doses of cocaine (20 and 40 mg/kg) were increased but not significantly different between WT and HET mice [9,47]. Importantly, in the present data, the exacerbated response to acute cocaine 10 mg/kg was higher in HET female than HET male rats. In contrast, when a higher dose of cocaine is presented acutely (20 mg/kg), male HET rats responded for a more sustained period than females. 

Although addiction affects both sexes, it develops and progresses differently in females compared with males [21]. Women report a greater subjective high in response to cocaine, even when drug levels and metabolite production are equivalent across sexes [22,23]. The abundant literature shows that there are significant sex differences in both acute responses to stimulant drugs and in adaptations to repeated drug use [25,27,53]. Both male and female HET rats showed increased responses to cocaine administration (at both doses evaluated) when compared to WT rats, however, our data confirms more pronounced effects of cocaine, across different exposures (lower dose acute, chronic and sensitization conditions) in female rats. After repeated exposures to the same cocaine dose (chronic condition), we found a stronger impact of cocaine (compared to acute condition), in HET female rats, pointing out the relevance of sex as well as the partial mutation of DAT in mediating cocaine behavioral outcomes.

Cocaine users and addicts go through some periods of “washout” in their efforts to detoxify. However, chronic exposure to cocaine induces neuroadaptations that prime the system to a new exposure. In accordance with our understanding of the role of DAT mutations to mediate vulnerability to relapse, we tested DAT HET and KO rats in response to an acute cocaine dose after a period of washout. When a challenge was presented seven days after the last cocaine exposure (sensitization condition), no effects of cocaine were detected in WT males. However, the effects were magnified in WT females and sustained for a longer period in HET females, indicating augmented response to cocaine (same dose as previously exposed, 10 mg/kg). 

The DA system is the most important mediator of cocaine’s addictive properties, especially via DAT [54]. Ji et al. [55] previously reported more robust effects of DAT mutations in DAT HET female striatal DA concentration and function compared to male mice. Recently, Sanna et al. [56] reported, in males, an approximate 3-fold increase in Nucleus Accumbens (Nacb) DA in HET DAT rats, in comparison to WT, while the increase in full KO DAT rats was close to 10-fold. However, because cocaine exerts its function via DAT but also other mechanisms, such as serotonin [57,58], precautions need to be taken when interpreting the results obtained with DAT KO mice and rats; other underlying mechanisms should be considered. Previous studies in DAT KO mice [45] concluded that cocaine increases extracellular DA independently of DAT. We decided to study, via protein analyses, the underlying mechanisms related to DA that could be playing a role on the behavioral outcomes in response to cocaine. It is important to consider changes at a secondary level as responses to the targeted mutation in the DAT gene, which could modify DA synthesis, transport or even degradation, altogether exerting an impact on behavioral outcomes when exposed to drugs of abuse. 

TH and VMAT2 are main regulators of DA and play important roles in mediating DAT activity. TH is activated to make DOPA, which after decarboxylation to DA is transferred into the vesicle by VMAT2. VMAT2 co-localizes with TH to build a protein complex that ensures DA is packaged promptly into secretory vesicles rather than remaining free in the cytosol [59]. Previous publications with DAT KO mice differ in the outcomes of DA levels, therefore, different behavioral responses have been found when exposed to a self-administration cocaine paradigm [45,60]. Those experiments raised the flag regarding the need for a better understanding of DA mediators in those animals for their important roles on DA levels and related role-mediating cocaine response. Alterations in TH and VMAT can play a crucial role in mediating DA levels in these animals. The present study highlighted important alterations in those modulators of DA levels and activity. The expression of TH protein in KO rats in striatum was greatly diminished compared to WT rats, but no changes were found in HET rats. Previous publications also reported decreases of TH in striatum and Nacb of DAT KO mice [12,61,62]. However, gene levels of *TH* did not show significant changes in KO DAT male rats in ventral striatum compared to their WT controls [63]. Data with DAT HET mice differ between studies based on the TH phosphorylation sites studied in the striatum; a few studies [61,62] did not report TH changes in HET mice (pSer40), but Jones et al. [12] reported a significant decrease in those mice (not phosphorylation site reported). Leo et al. [19] showed reduced *TH* RNA levels in the midbrains of mixed group of males and female KO rats and a reduction tendency in HET rats that did not reach statistical significance. Our study is the first one to report TH protein levels in DAT HET and KO male and female rats separately. In line with results from Leo et al. (2018), male and female DAT rats showed reduced TH protein levels only in full KO rats. In DAT KO mice [9], as well as in DAT KO rats [19], a downregulation of striatal presynaptic D2 receptors have been described. Due to the role of presynaptic D2 receptors inhibiting TH, D2 receptor reduction could give support to DA increases at extracellular levels, enhancing TH activity to synthesize DA [12] even when TH levels are reduced. Intact striatal levels of TH in HET DAT rats indicate that there are other main targets involved in the DAT-DA system modulating the increased vulnerability that HET rats present when exposed to cocaine. 

VMAT2 was also linked to the effects of psychostimulants and, therefore, suggested as a target for the treatment of cocaine abuse [64]. VMAT2′s role regulating cytosolic concentrations and availability of DA showed to also be compromised in DAT male and female rats. Previous studies showed parallel effects between VMAT2 and DAT in KO mice [65]. There were also no significant alterations at a protein or mRNA level in striatum. Neither SNpc was detected in DAT HET male or female mice [55]. However, for the first time, our study showed reduced VMAT2 protein levels in the striatum of male and female HET and KO DAT rats. Although genetic *VMAT2* deficiencies have been previously linked to decreases in the synaptic release of monoamines [66], our study suggests that although the protein levels of TH and VMAT2 are reduced in our rats, TH and VMAT stay functional and, due to the lack of DAT feedback signal, could contribute to an increased release of extracellular DA. Previous studies [55] reported that the function of DAT and VMAT may be linked and play different roles regulating extracellular DA concentrations (DAT’s direct role) versus direct cytosolic DA concentrations (VMAT’s main role) with VMAT only having indirect effects on extracellular DA. Both transporters play a critical role in dopaminergic function by regulating the availability and access of intracellular and extracellular DA. The reduction of VMAT2 in mice models has been linked to increased responses to cocaine [65,67]. Further studies are needed for a more detailed mechanistic understanding. VMAT alterations, reducing DA pool at similar levels in males and females, as well as in HET and KO DAT rats, do not seem to be sufficient to explain the behavioral outcomes drastically different between HET and KO DAT rats.

There is strong evidence supporting the role of α-Syn presynaptically mediating DA levels. However, there is contradictory data about the direction of those effects [68]. α-Syn plays an important role in DA concentration at nerve terminals via action on multiple steps of the DA metabolism [69], such as DA synthesis, storage, release, and re-uptake [70,71]. Additionally, α-Syn co-localizes with TH [72] and influences its activity. Previous studies highlight the increased levels of α-Syn in humans after repeated cocaine exposure [4,73]. Perez et al. [72] stated that a reduction in expression or aggregation of α-Syn would directly impact DA and increase its synthesis. One additional explanation for the effects of α-Syn on the DA system is via α-Syn effects decreasing DAT [74]. A recent study highlighted the role that sex and α-Syn can have in responses to cocaine; cocaine only increased exosome α-Syn in female brains but not in males, effects that were neutralized by ovariectomy [75]. In our study, we found reduced α-Syn monomer in HET and KO female rats but no significant effects in HET males. This sex difference in the HET group of rats could give additional support to the increased effect of small doses of acute cocaine as well as chronic exposure in female HET compared to male HET rats, and could highlight the relevance of hormone levels in vulnerability and behavioral response to cocaine. 

In addition to these long-known targets related to DA transmission, we studied a new target involved in mediating DAT function and/or expression. In DAergic neurons, protein interacting with C Kinase-1 (PICK1) co-localizes with DAT and forms a stable protein complex [76]. Importantly, PICK1 can be found at pre- and postsynaptic sites in the cortex, hippocampus, cerebellum, and striatum [77,78]. PICK1 interacts with PKCa to mediate contacts between PKCa and membrane proteins in the CNS [79] and plays an important role in synaptic plasticity [80]. The DAT-PICK association causes an increase in cell-surface DAT expression and increases DAT uptake capacity [76], suggesting an important role for PICK1 in DAT expression and its role mediating the response to stimulants, and highlights its possible importance as a therapeutic target. Jensen et al. [81] studied the response to cocaine in PICK KO mice and found it to be playing a relevant role on cocaine locomotor-induced properties. Our study links reductions in striatal PICK protein levels with increased responses to cocaine in HET rats and provides additional information regarding PICK1 as an innovative target for the treatment of cocaine abuse and addiction. PICK1 protein levels were more abruptly reduced in female HET and KO DAT rats that in males, when compared to their same sex WT controls. Recently, Wickens et al. [82], highlighted relevant sex differences regarding alterations of PICK levels and their implication on the response to cocaine seeking (addictive) behavior. In line with our current results showing sex differences in PICK1–cocaine mediation, disruption of PICK (conditional deletion) increased cocaine seeking only in females, while it had opposite effects in male mice. Further studies are encouraged to better understand the role of PICK in addiction. Hormone role in response to drugs of abuse has previously been described and cannot be excluded from sex differences in response to drugs of abuse. Future studies will focus on the role of those proteins in response to cocaine.

Our data suggest that changes in direct mediators of DA synthesis, transport, and release, in combination with diminished DAT expression, are critical to mediating predisposition to drug abuse and could play a key role in mediating behavioral outcomes related to drugs of abuse and in changes associated with chronic drug use.

Altogether, these data highlight the increased vulnerability to drugs of abuse in females compared to males, especially after repeated exposure during adolescence. As discussed previously, sex is an important variable that influences the response to drugs of abuse. However, additional information is needed regarding DAT’s role and the impact that genetic DAT perturbation can have on several dopaminergic functions based on sex differences [55]. Importantly, the rate of comorbidity between drug abuse and anxiety disorders is higher in women than men [83], which gives support to the relation between our current results and our previous study demonstrating that HET female rats are more vulnerable to stress effects [51].

## 4. Materials and Methods

### 4.1. Drugs

Cocaine hydrochloride was obtained from the National Institute on Drug Abuse (Rockville, MD, USA). Cocaine was administered intraperitoneally (ip). Cocaine doses (0, 10 or 20 mg/kg) were chosen based on previous publications [84,85,86]. Saline 0.9% solution was used as vehicle.

### 4.2. Animals

*Slc6a3* (DAT) KO rats were generated previously [19] and kindly provided by Dr. Gainetdinov. The DAT-KO rat colony was kept under HET-HET breeding at our facilities following the expected Mendelian ratio. A total of 68 males and 71 females were used (males: WT N = 15, HET N = 36, KO N = 17; females: WT N = 24, HET N = 24, KO N = 23). Rats were genotyped at postnatal day (PND) 18–21 as previously described [19]. We recently confirmed reduced striatal protein DAT levels in HET as well as lack of DAT in KO rats [51], Animals were weaned at PND 21–23. Animals were housed in groups after weaning in a temperature (21 ± 1 °C) and humidity-controlled environment under a 12 h light/dark cycle. Animals from the three different genotypes (wild-type (WT), heterozygous (HET) and homozygous (KO)) were used. The study was performed with rats aged postnatal day (PND) 35 to PND48, the period considered as adolescence in rats [87,88,89]. Between experimental sessions, the rats had continuous access to water and food in their home cages. The studies were conducted in accordance with the guidelines of the Guide for the Care and Use of Laboratory Animals (Eighth Edition), National Research Council, Department of Health, Education and Welfare (2011) and all studies were approved by the University of Miami Institutional Animal Care and Use Committee. 

### 4.3. Immunoblot Analysis

At PND42, a subgroup of rats was killed by decapitation 60 min after saline treatment (N = 4–5 per genotyping group). Rat striatum was dissected and was mechanically homogenized in a RIPA buffer (#89901, Pierce, Waltham, MA, USA) plus protease inhibitor mixture (#1861280 Thermo Scientific, Waltham, MA, USA). Protein concentration was measured using a BCA protein assay kit (#23227 Thermo Scientific). Protein extracts (20 ug) were separated by 10% SDS/PAGE and transferred to nitrocellulose membranes. Blots were incubated overnight at 4 °C with the following primary antibodies: tyrosine hydroxylase (TH) (#AB5935; Millipore, Toronto, ON, Canada); vesicular monoamine transporter 2 (VMAT2) (#ab70808); α-Synuclein (αSyn) (#610787; BD Transduction Laboratories, San Jose, CA, USA); PICK1 (#ab3420; Abcam, Cambridge, UK); GAPDH (#ab181602; Abcam). After washing, the membranes were incubated for 1 h at room temperature with the appropriate secondary antibody (Rdye anti-mouse #926-68070 or Rdye anti-rabbit #926-32211). Where needed, the membranes were stripped for 10 min with Restore™ PLUS Western Blot stripping buffer (Thermo Scientific #46430), blocked, and incubated with primary antibody, as described above. After washing in PBS-Tween20 0.05%, the membranes were incubated for one hour at room temperature with the appropriate Horseradish peroxidase (HRP) conjugated secondary antibody (ECL™ Anti-Rabbit IgG Sigma #NA9340V; ECL™ Anti-Rabbit IgG Sigma #NA931, 1:2000 in PBS-Tween20 0.05%). Following secondary antibody incubations, the membranes were washed and incubated with ECL detection reagent (Thermo Fisher #34578) for 1–2 minutes. Immunoblots were acquired while using the Bio Rad ChemiDoc™ Touch systems (Bio Rad, USA) and images were quantified using Image Lab 5.2.1 software (Bio Rad, USA). For each protein, the data were normalized to GAPDH and expressed as relative to control (WT). After secondary antibody incubations, the membranes were washed and finally visualized using LICOR imager (LI-COR Odyssey, USA). Images were quantified using Image Studio Lite Ver 5.2 online software. Bands were normalized for housekeeping gene GAPDH.

### 4.4. Open Field (OF) Apparatus

Rats were placed in clear acrylic chambers (16 × 16 inches) inside Digiscan activity monitors (Omnitech Electronics, Columbus, OH) that were equipped with infrared light sensitive detectors mounted 2.5 cm apart along two perpendicular walls. Mounted along the opposing walls were infrared light beams that were directed at the detectors. One count of horizontal activity was registered each time the subject interrupted a beam, and the distance traveled was calculated automatically based upon successive light beam breaks. Animals were maintained on a 12 h light/dark schedule with lights on at 7 a.m. and off at 7 p.m. All behavioral testing was completed during the light schedule between 9 a.m. and 2 p.m. with groups randomized over the course of the day.

Locomotor activity testing. 24 h prior to testing, rats were habituated to handling and saline injections. Animals were tested during the adolescent developmental stage. At postnatal day PND 35 (acute condition), rats were placed into the OF for a period of 30 min. The first 10 min were analyzed for a measure of novelty and the last 10 min of this period were analyzed as a baseline measure of habituation. At min 30, for acute testing of cocaine’s effects, all rats were injected intraperitoneally (ip) with vehicle (saline), or cocaine (10 or 20 mg/kg), and locomotor activity was recorded for an additional 60 min. After testing, animals were returned to their home cages and, for the study of chronic effects cocaine, rats were injected daily with the same dose of cocaine or saline from PND36 to PND41. Seven days of 10 mg/kg cocaine was chosen based on our preliminary data and previous studies performed in Sprague Dawley rats showing that seven days of 10 mg/kg cocaine is enough to develop sensitization to its locomotor effects [85]. On PND42 (chronic condition), the same procedure as described for acute condition was repeated (rats were again injected and tested in the open field chamber for locomotor activity). After testing, animals were returned to their home cages and left undisturbed until PND48. On PND48 (sensitization condition), rats were again injected with drug or saline (same dose of cocaine or saline) and tested for locomotor activity (as previously described). 

### 4.5. Data Analyses

The distance traveled (cm) data were analyzed using a three-way ANOVA (genotype x sex x dose of cocaine) for the acute data. For the chronic study, a four-way ANOVA (genotype x sex x dose x test day), followed by Tukey’s HSD, where appropriate. SPSS was used for these analyses. Protein levels were analyzed by a one-way ANOVA (genotype) using GraphPad PRISM6 software. All data were expressed as mean ± SEM, and significance was set at *p* < 0.05.

## 5. Conclusions

We conclude, in agreement with the previous literature, that behavioral response to cocaine depends on DAT expression level and related increases in extracellular DA [90]. However, this study is innovative because it uses DAT transgenic rats, for the first time, for the study of the effects of cocaine. Rats are more similar to humans in their genetics and pharmacokinetics than mice [91,92] and present greater synaptic complexity [85], suggesting that results of therapeutics tested in rats would provide greater predictive validity than testing in mice. The rat is being proposed as an ideal model due to their ability to perform more complex tasks than mice. The use of transgenic rats is getting more popular and is already being used in studying a variety of medical conditions (such as neurological disorders including Alzheimer and Huntington’s Disease as well ALS, between others). Differences found between DAT HET and KO rats expand upon previous publications in DAT KO mice and highlight the impact of DAT levels in locomotor activity as well as in response to cocaine. Additionally, sex differences raised in the current study highlight the need for further studies on dopaminergic signaling in response to cocaine, as well as the role of sexual hormones. The adaptations seen in HET rats at a neurochemical level differ from those seen in full KO rats and highlight the importance of studying HET rats as a useful tool for the study of DAT-related disorders, such as cocaine addiction. We would like to point out that several human diseases have been linked to a reduction or polymorphism of DAT but not to a full DAT KO. Dysregulation of DAT has been described, including Parkinson’s Disease [93,94], ADHD [95], PTSD [96,97] bipolar disorder [98], Excited Delirium Syndrome [99] and obesity [100]. This new rat model represents an innovative genetic rat that will help the development of new target strategies for drug addiction, but not limited to it.

## Figures and Tables

**Figure 1 ijms-23-15414-f001:**
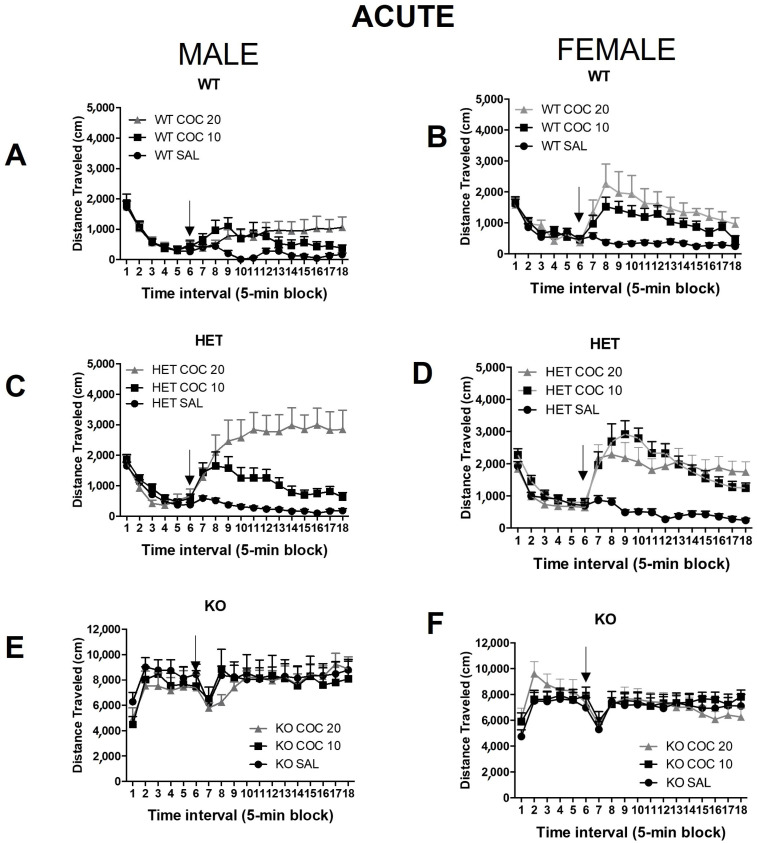
Acute effects of cocaine on locomotor activity in DAT mutant rats. At PND35, rats were habituated to the locomotor test chamber for 30 min. Cocaine (0, 10 or 20 mg/kg) was injected (ip) and rats were returned to the chamber and monitored for 60 more min. Time course of locomotor activity of male (**A**) Wild-type (WT), (**C**) Heterozygous (HET) DAT rat and (**E**) Homozygous (KO) DAT Knockout rats, and female (**B**) Wild-type (WT), (**D**) Heterozygous (HET) DAT rat and (**F**) Homozygous (KO) DAT Knockout rats. Arrow indicates the time of treatment. Mean ± SEM. (males: WT N = 15, HET N = 36, KO N = 17; females: WT N = 24, HET N = 24, KO N = 23).

**Figure 2 ijms-23-15414-f002:**
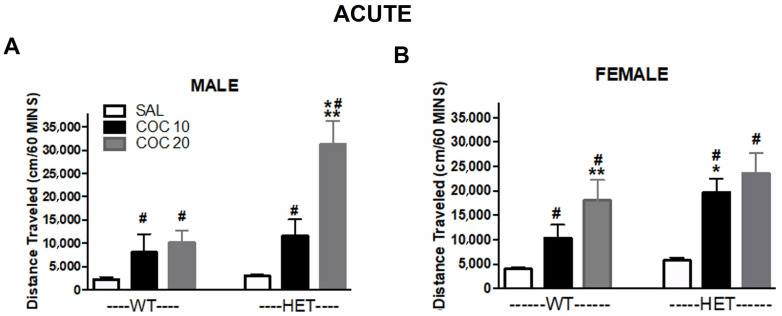
Effect of cocaine on cumulative distance traveled in DAT mutant rats. Cumulative time (30 min after acute treatment) is represented across conditions (SAL = saline, COC 10= 10 mg/kg cocaine; COC 20= 20 mg/kg cocaine) in males (**A**) and females (**B**). # *p* < 0.05 compared to SAL; * *p* < 0.05 compared to WT; ** *p* < 0.05 compared to 10 mg/kg. Mean ± SEM. (males: WT N = 15, HET N = 36; females: WT N = 24, HET N = 24).

**Figure 3 ijms-23-15414-f003:**
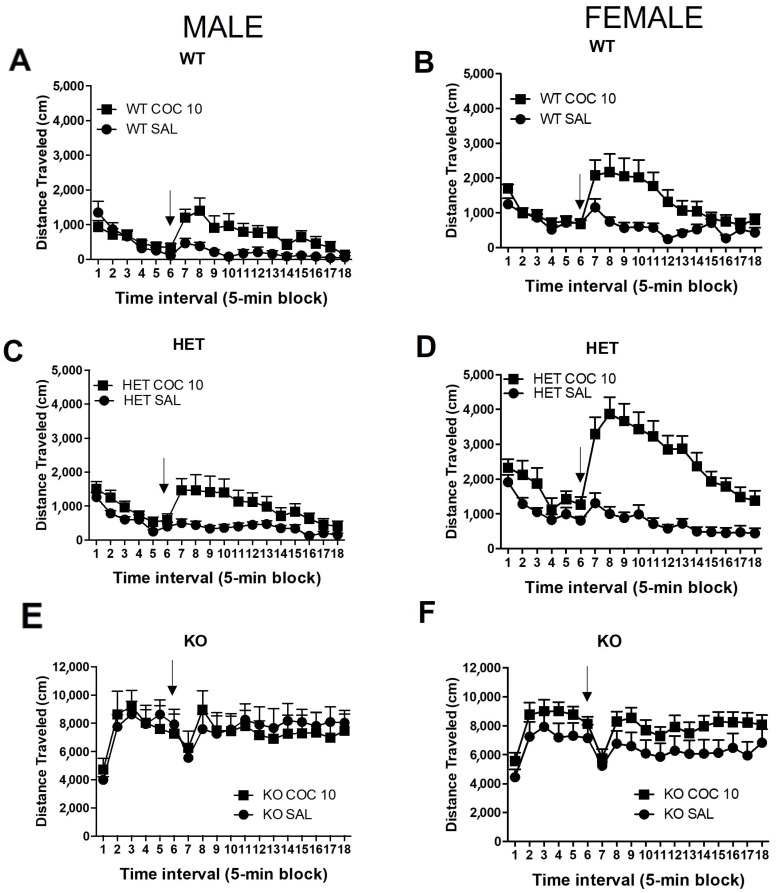
Chronic effect of cocaine on locomotor activity of DAT mutant rats. At PND42, and after repeated daily cocaine injections in the home cage (PND36 to PND41), rats were habituated to the locomotor test chamber for 30 min. Cocaine (0 or 10 mg/kg) was injected (ip) and rats were returned to the chamber and monitored for 60 more min. Time course of locomotor activity of: male (**A**) Wild-type (WT), (**C**) Heterozygous (HET) DAT rat and (**E**) Homozygous (KO) DAT Knockout rats, and female (**B**) Wild-type (WT), (**D**) Heterozygous (HET) DAT rats, and (**F**) Homozygous (KO) DAT Knockout rats. Arrow indicates the time of treatment. Mean ± SEM. (males: WT N = 15, HET N = 36, KO N = 17; females: WT N = 24, HET N = 24, KO N = 23).

**Figure 4 ijms-23-15414-f004:**
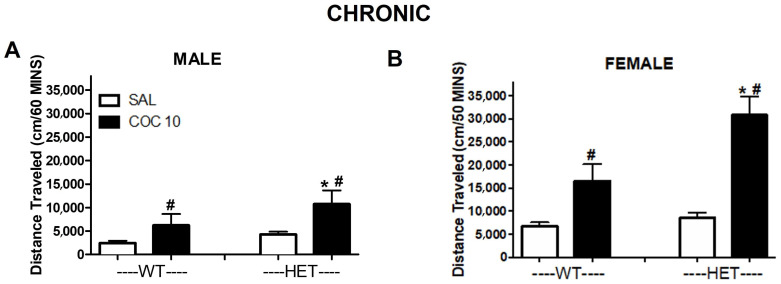
Effect of chronic cocaine (10 mg/kg) on cumulative distance traveled in DAT mutant rats. Cumulative time (30 min after acute treatment) is represented across conditions (SAL = saline, COC 10= 10 mg/kg cocaine) in males (**A**) and females (**B**). # *p* < 0.05 compared to SAL; * *p* < 0.05 compared to WT. Mean ± SEM. (males: WT N = 15, HET N = 36; females: WT N = 24, HET N = 24).

**Figure 5 ijms-23-15414-f005:**
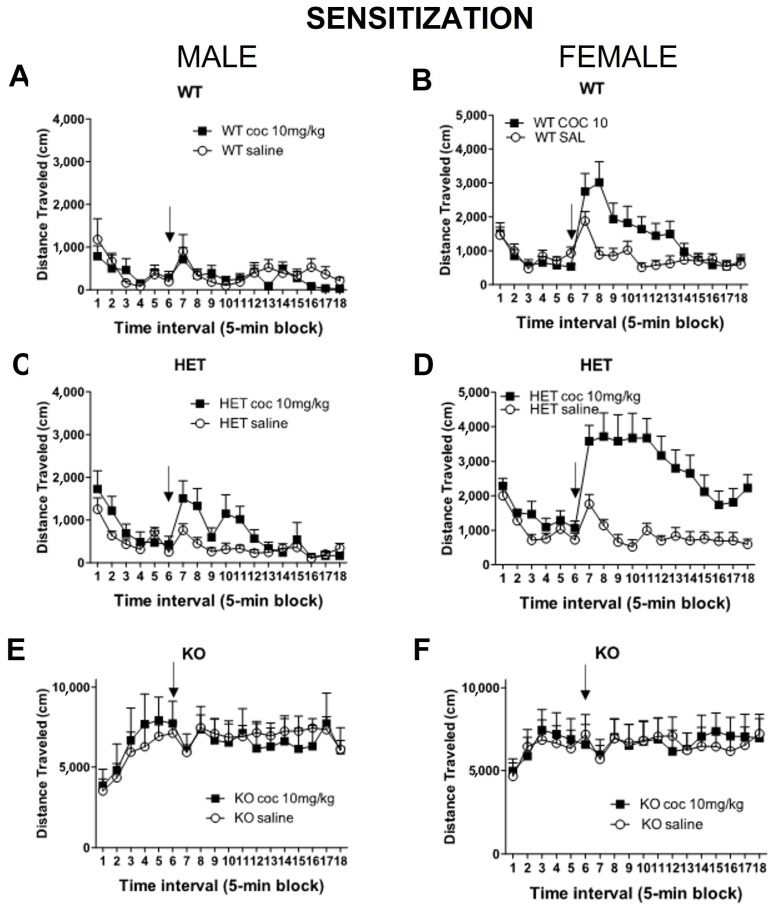
Effect of cocaine (sensitization challenge, 10 mg/kg) on locomotor activity in DAT mutant rats. At PND42, and after repeated daily cocaine injections in the home cage (PND36 to PND41), rats were left undisturbed in their home cages. At PND48, rats were habituated to the locomotor test chamber for 30 min. Cocaine (0 or 10 mg/kg) was injected (ip) and rats were returned to the chamber and monitored for 60 more min. Time course of locomotor activity of male (**A**) Wild-type (WT), (**C**) Heterozygous (HET) DAT rat and (**E**) Homozygous (KO) DAT Knockout rats, and female (**B**) Wild-type (WT), (**D**) Heterozygous (HET) DAT rat and (**F**) Homozygous (KO) DAT Knockout rats. Arrow indicates time of treatment. Mean ± SEM. (males: WT N = 15, HET N = 36, KO N = 17; females: WT N = 24, HET N = 24, KO N = 23).

**Figure 6 ijms-23-15414-f006:**
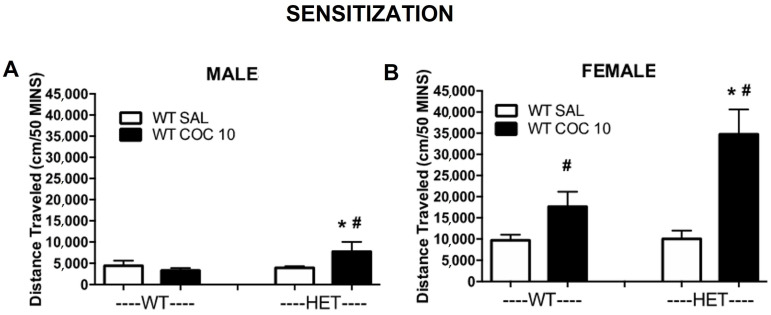
Effect of cocaine (sensitization challenge, 10 mg/kg) on cumulative distance traveled in DAT mutant rats. Cumulative time (30 min after acute treatment) is represented across conditions (SAL = saline, COC 10= 10 mg/kg cocaine) in males (**A**) and females (**B**). # *p* < 0.05 compared to SAL; * *p* < 0.05 compared to WT. Mean ± SEM. (males: WT N = 15, HET N = 36; females: WT N = 24, HET N = 24).

**Figure 7 ijms-23-15414-f007:**
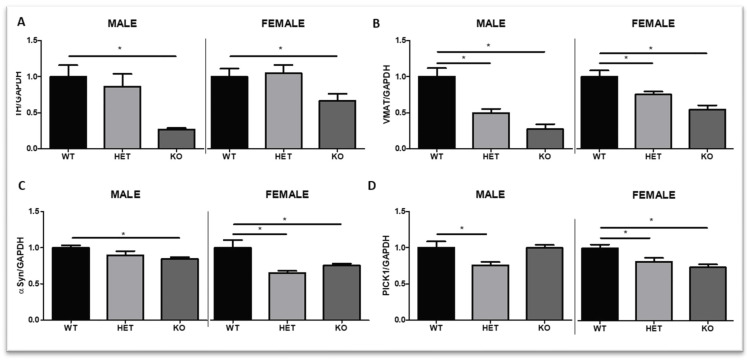
Striatal protein expression levels of (**A**) Tyrosine-hydroxylase (TH), (**B**) Vesicular monoamine transporter 2 (VMAT2), (**C**) α-Synuclein (α-Syn) and (**D**) protein interacting with C-kinase-1 (PICK) in DAT mutant rats, at PND42. Quantitative analysis. * *p* < 0.05 compared to WT rats. Mean ± SEM. N = 4–5 per group.

## Data Availability

Data is contained within the article and Appendix A.

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
