# Peer review of "Heterozygote Dopamine Transporter Knockout Rats Display Enhanced Cocaine Locomotion in Adolescent Females"

_ijms, 2022, doi:10.3390/ijms232315414_

Round 1
Reviewer 1 Report
This is a very well written paper on the effect of cocaine addiction in DAT HET rat adolescents. I have just one time technical question.
Why did you placed the Methods section following the Discussion section and not the Introduction section?
Author Response
Why did you placed the Methods section following the Discussion section and not the Introduction section?
We thank and appreciate the nice comments of the reviewer. We acknowledge that having the methods before the results may help the understanding of the results, however, we followed the journal template. IJMS provides a template to follow when preparing the manuscript. On that template the methods are located at the end of the document.

Reviewer 2 Report
Could the authors provide 'n' for data in each figure.
Fig 2 cummulative data - 30 min is used - is this related to the expected peak effects of cocaine in rats or half-life?
Methods - did you test female rats' hormone levels to assess cycle? could time in cycle affect cocaine induced locomotion?
Discussion - you mention limitation of mice models (line 346) - can you expand?
Discussion - i can understand that TH is reduced as with no DAT, the need to make more DA is presumably reduced. I don't understand why levels of vMAT2 decrease as i'd have thought sequestering DA might be useful in a hyperdopaminergic state. Can the authors comment?
there are small grammar/English issues throughout
Author Response
We appreciate reviewer’s comments
- Could the authors provide 'n' for data in each figure.
Per reviewer request, we have included the N on the figure legends.
-F2 cummulative data-30min is used- is this related to the expected peak effects of cocaine in rats or half-life?
We thank reviewer’s comment. We observed that the peak effect of cocaine under all doses tested was between 10-20min after IP administration. Due to that and the previously described half-life of cocaine in rats being of approximately 30min, we decided to analyze the first 60 min after administration (as shown in the time course figures) but in the cumulative results, to highlight main differences between groups, we represent the time frame 0 to 30 min after cocaine administration, what is representative of the peak effects. Our results are in agreement with previous publications showing peak response to intraperitoneal cocaine before 30 min (see for example Gulley et al., 2003)
Gulley JM, Hoover BR, Larson GA, Zahniser NR. Individual differences in cocaine-induced locomotor activity in rats: behavioral characteristics, cocaine pharmacokinetics, and the dopamine transporter. Neuropsychopharmacology. 2003 Dec;28(12):2089-101. doi: 10.1038/sj.npp.1300279. PMID: 12902997.
-Methods:
Did you test Female rat's hormone levels?
We acknowledge the relevant role that sex hormones have on sex differences. However, we did not focus this work on sex hormones and we did not measure it. We appreciate the comment and we will be sure to include hormone levels in our future studies. We briefly discuss the implication of hormones on the parameters measured. In our discussion we mention the relevance of sex hormone on the possible explanation of our results and make emphasis on their relevance to measure them in future studies. We cited the following:
“This sex difference in the HET group of rats could give additional support to the increased effect of small doses of acute cocaine as well as chronic exposure in female HET compared to male HET rats and could highlight the relevance of hormone levels in vulnerability and behavioral response to cocaine. “
“Hormone role in response to drugs of abuse has previously been described and cannot be excluded from sex differences in response to drugs of abuse.”
“sex differences raised in the current study highlight the need of further studies on dopaminergic signaling in response to cocaine as well as the role of sexual hormones.”
Could time in cycle affect cocaine induced locomotion?
Yes, estrogen has been shown to have strong effects mediating effects of cocaine.
Our study does not focus on the effects of estrogen on the sex differences found, however, we mention its relevance as previously explained; sex hormones could be underlying some of the changes found in behavioral parameters as well as on protein levels.
A growing body of evidence implicates the steroid hormone estradiol in mediating this sex difference and other research articles previously focused on the effects of several hormones mediating sex differences in the addictive properties of cocaine (see for example Martinez et al., 2014)
Martinez LA, Peterson BM, Meisel RL, Mermelstein PG. Estradiol facilitation of cocaine-induced locomotor sensitization in female rats requires activation of mGluR5. Behav Brain Res. 2014 Sep 1;271:39-42. doi: 10.1016/j.bbr.2014.05.052. Epub 2014 Jun 2. PMID: 24893316; PMCID: PMC4636080.
-Discussion:
You mention limitation of mice models (line 346)-can you expand?
We thank reviewer’s comment. We acknowledge the relevance of published mice data to study drug addiction. Mice models, for decades, have provided strong evidence to increase the understanding of the underlying mechanisms mediating cocaine effects. Additionally, lots of mice studies have tried to explain the addiction process. However, unfortunately, there is no current FDA approved treatment for cocaine addiction. We are currently facing what some authors call the “translational failure”, where some mice research is poorly reproducible and often does not replicate in humans.
Brian et al., (2021) acknowledged that no one animal model can fully recapitulate the complexity of a human disease. However, they also highlight the morphologic and functional similarities from basic mammals and humans. But the higher size the mammal has, greater homology at a gene, cell, tissue and organ system levels seem to occur.
(Brian R Berridge, Animal Study Translation: The Other Reproducibility Challenge, ILAR Journal, Volume 62, Issue 1-2, 2021, Pages 1–6, https://doi.org/10.1093/ilar/ilac005)
As murine research becomes increasingly sophisticated, researchers are quickly realizing that mice have some significant limitations as a research model. As we highlight in our discussion, rats are larger in size than mice, but they are also physiologically and genetically more similar to humans, which, according to researchers, allows for transgenic rat models to better replicate the spectrum of human pathology.
Additionally, Snyder et al., (2016) indicated that advantages in behavioral testing may provide the greatest motivation for making the switch to rats as a research model. As briefly mentioned in our discussion, “Cognitive scientists in general are increasingly embracing the rat as an ideal behavioral test subject for a broad spectrum of behavioral studies, arguing that they are capable of more complex tasks than mice but with greater potential for genetic manipulation than nonhuman primates (Abbott, 2004)”
Transgenic rats have the ability to be effectively used across a wide variety of research applications and are already being used in sarcomas, metabolic syndromes, neurological disorders (f.e. Alzheimer’s Disease and Huntington’s Disease), HIV and Amyotrophic Lateral Sclerosis (ALS). These new transgenic rats are proposed to allow better understanding of human disease pathology. Our article is the first one to introduce this new transgenic model as a new animal model for the study of cocaine addiction.
(Snyder, Jason S., et al. “A Transgenic Rat for Specifically Inhibiting Adult Neurogenesis.” Eneuro, vol. 3, no. 3, 2016, https://doi.org/10.1523/eneuro.0064-16.2016.)
(Abbott A (2004) Laboratory animals: the Renaissance rat. Nature 428:464–466. 10.1038/428464a )
We briefly explained in our conclusion the advantages of these rats in comparison to transgenic mice, as follows: “ Rats are more similar to humans in their genetics and pharmacokinetics than mice [83,84] and present greater synaptic complexity [85] suggesting that results of therapeutics tested in rats would provide greater predictive validity than testing in mice. Differences found between DAT HET and KO rats expand previous publications in DAT KO mice and highlight the impact of DAT levels in locomotor activity as well as in response to cocaine.”
However, per reviewer suggestion, we have modified the text to make more emphasis on the limitations of mice models or more precisely, the benefits of new transgenic rats, as follows in line 646: “ Rats are more similar to humans in their genetics and pharmacokinetics than mice [83,84] and present greater synaptic complexity [85] suggesting that results of therapeutics tested in rats would provide greater predictive validity than testing in mice. The rat is being proposed as an ideal model due to their ability to perform more complex tasks than mice. The use of transgenic rats is getting more popular and is already being used in a big variety of medical conditions (such as neurological disorders including Alzheimer and Huntington’s Disease as well ALS, between others).”
-I can understand that TH is reduced as with no DAT, the need to make more DA is presumably reduced. I don’t understand why levels of VMAT2 decrease as I’d have thought sequestering DA might be useful in a hyperdopaminergic state. Can the authors comment?
We would like to thank reviewer’s observation. We acknowledge the complexity of the mechanisms underlying DA release and the possible alterations that the transgenic model could have as a consequence of DAT reduction or full depletion.
One of the possible explanations of decreased VMAT2 on HET and KO DAT rats could be neuroadaptations trying to reduce the DA that arrives to the presynaptic terminal.
Due to the fact that these rats present a hyperdopaminergic state, a reduction on TH as well as VMAT2 could be an internal adaptation to search for DA homeostasis. Our data agrees with Miller et al. (2001 [65]) who described reduced DAT in VMAT2 heterozygous mice. Miller suggests that therapies resulting in increased vesicular uptake by VMAT2 may act to decrease cytosolic dopamine [65].
However, as we describe in our discussion, “Even though genetic VMAT2 deficiency have been previously linked to decreases in the synaptic release of monoamines [66], our study suggests that although protein levels of TH and VMAT2 are reduced in our rats, TH and VMAT stay functional and, due to the lack of DAT feedback signal, could contribute to increase release of extracellular DA. Further studies are needed for a more detailed mechanistic understanding. VMAT alterations, reducing DA pool at similar levels in males and females, as well as in HET and KO DAT rats, do not seem to be sufficient to explain, in our study, the behavioral outcomes drastically different between HET and KO DAT rats”.
Transgenic models help elucidate the role of main genes. However, since the genetic modification is already present at an embryonic stage, other alterations in the DA mechanisms could develop, such as those observed on VMAT2 protein levels. However, the reduction on VMAT has previously been linked to increased response to cocaine (Miller et al., 2001; Wang et al., 1997). With only 50% of normal VMAT2, VMAT heterozygote mice have reduced vesicular filling and DA release and these alterations in presynaptic monoamine function in the heterozygotes are thought to be responsible for the observed sensitization to the psychostimulants such as cocaine (Miller et al., 2001)
(Miller GW, Wang YM, Gainetdinov RR, Caron MG. Dopamine transporter and vesicular monoamine transporter knockout mice : implications for Parkinson's disease. Methods Mol Med. 2001;62:179-90. doi: 10.1385/1-59259-142-6:179. PMID: 21318776.)
(Wang, Y. M., Gainetdinov, R. R., Fumagalli, F., Xu, F., Jones, S. R., Bock, C. B., Miller, G. W., Wightman, R. M., & Caron, M. G. (1997). Knockout of the vesicular monoamine transporter 2 gene results in neonatal death and supersensitivity to cocaine and amphetamine. Neuron, 19(6), 1285–1296. https://doi.org/10.1016/s0896-6273(00)80419-5)
Ji et al., (2009) reports that the function of DAT and VMAT may be linked and play different roles regulating extracellular DA concentrations (DAT’s direct role) versus direct cytosolic DA concentrations (VMAT’s main role) with VMAT only having indirect effects on extracellular DA. Both transporters play a critical role in dopaminergic function by regulating availability and access of intracellular and extracellular DA.
(Ji, J., M. Bourque, T. Di Paolo and D. E. Dluzen (2009). "Genetic alteration in the dopamine transporter differentially affects male and female nigrostriatal transporter systems." Biochem Pharmacol 78(11): 1401-1411.)
To better explain the results obtained, as just described, as suggested by reviewer, we have modified the discussion of VMAT2 results as follows in line 503: “. Previous studies [55] reported that the function of DAT and VMAT may be linked and play different roles regulating extracellular DA concentrations (DAT’s direct role) versus direct cytosolic DA concentrations (VMAT’s main role) with VMAT only having indirect effects on extracellular DA. Both transporters play a critical role in dopaminergic function by regulating availability and access of intracellular and extracellular DA. The reduction of VMAT2 in mice models has been linked to increased response to cocaine [65,101].”
Small grammar/English issues throughout
As per reviewer suggestion, we have performed a deep English revision of the manuscript. Track changes are used on the manuscript for each modification done.

Reviewer 3 Report
This article describes the effect of cocaine in a new dopaminergic transgenic rats model. The manuscript is quite well written. The figures are clear, and the results are interesting. The work represents a contribution to the knowledge of addictions. I suggest accepting it after the following minor considerations:
Introduction
- In this section, the author says "...Major psychostimulants, such as cocaine and amphetamine...". Since this journal belongs to "...Molecular Science" I suggest including a figure in the introduction with the chemical structures of these psychostimulants and their IC50 values ( or KI, or Kd) in the DAT.
Results
- In various parts of the manuscript the acronym PND35 is mentioned. I suggest defining "PND", so that it be clear to non-specialized readers.
Materials and methods
- page 16, line 560: "...saline..." is mentioned. I suggest specifying the concentration used.
- page 16, line 585: correct 4Cº
Conclusions
Page 17, line 639 "We conclude, in agreement with previous literature" Add reference to the previous literature to which they refer.
References
- It seems that references 94-99 do not have the required format. Please revise.
Author Response
Introduction- In this section, the author says "Major psychostimulants, such as cocaine and amphetamine..." Since this journal belongs to "…..Molecular Science" I suggest including a figure in the introduction with the chemical structures of these psychostimulants and their IC50 values ( or KI, or Kd) in the DAT.
We thank the reviewer for the suggestion. We acknowledge the journal belongs to the molecular science field, but the chemical structures of cocaine and related interactions with DAT at that levels are out of the scope of the current study. Previous publications present the chemical structure of cocaine and/or dopamine transporter extensively (see for example: Ramanujapuram, 2006; Zou et al., 2017; Cheng et al., 2015). We acknowledge the relevance of their structures for their interaction (cocaine-DAT) but we do not focus on molecular aspects of their structures and we do not see added value into the current research article by adding the figure suggested. The main focus of the current research article is the behavioral effects of cocaine in rats with genetically altered DAT levels and, as a consequence of that mutation, the study of other relevant proteins that could be “readjusted” on the so-called “hyperdopaminergic” system (such as VMAT, TH, a-syn and PICK1). However, if the reviewer considers this figure as critical for the publication, we will be pleased to add a figure following the suggestion.
Ramanujapuram, S. (2006). Dopamine Uptake Inhibition Potency Fluctuations of Cocaine at the Dopamine Transporter (Master's thesis, Duquesne University). Retrieved from https://dsc.duq.edu/etd/1082
Zou, M. F., Cao, J., Abramyan, A. M., Kopajtic, T., Zanettini, C., Guthrie, D. A., Rais, R., Slusher, B. S., Shi, L., Loland, C. J., & Newman, A. H. (2017). Structure-Activity Relationship Studies on a Series of 3α-[Bis(4-fluorophenyl)methoxy]tropanes and 3α-[Bis(4-fluorophenyl)methylamino]tropanes As Novel Atypical Dopamine Transporter (DAT) Inhibitors for the Treatment of Cocaine Use Disorders. Journal of medicinal chemistry, 60(24), 10172–10187. https://doi.org/10.1021/acs.jmedchem.7b01454
Cheng, M. H., Block, E., Hu, F., Cobanoglu, M. C., Sorkin, A., & Bahar, I. (2015). Insights into the Modulation of Dopamine Transporter Function by Amphetamine, Orphenadrine, and Cocaine Binding. Frontiers in neurology, 6, 134. https://doi.org/10.3389/fneur.2015.00134
Results
- In various parts of the manuscript the acronym PND35 is mentioned. I suggest defining "PND", so that it be clear to non-specialized readers.
We thank reviewer’s comment. PND refers to postnatal day. We include in our Materials and methods section, in the animal subsection, the following: “The study was performed with rats aged PND35 to PND48, period considered as adolescence in rats [97,98, 99].” Due to reviewer’s suggestion, we have added “postnatal day” before PND in line 576 to clarify that we refer to postnatal age.
Materials and methods
-page 16, line 560: "...saline.." is mentioned. I Suggest specifying the concentration used.
Thank you for the observation. “Saline 0.9%” was specified in line 564
page 16, line 585: correct 4C°
We thank the observation. It has been modified to 4° C
Conclusions
Page 17, line 639 "We conclude, in agreement with previous literature" Add reference to the previous literature to which they refer.
As suggested by the reviewer, we acknowledge the need of a reference and we have added it into the text and the reference list. Reference added is [100]
- Siciliano CA, Jones SR. Cocaine Potency at the Dopamine Transporter Tracks Discrete Motivational States During Cocaine Self-Administration. Neuropsychopharmacology. 2017 Aug;42(9):1893-1904. doi: 10.1038/npp.2017.24. Epub 2017 Jan 31. PMID: 28139678; PMCID: PMC5520781
References
- It seems that references 94-99 do not have the required format. Please revise.
We thank the observation. References 94 to 99 have been properly formatted and introduced in the reference list as follows:
- Mandt, B. H., & Zahniser, N. R. (2010). Low and high cocaine locomotor responding male Sprague-Dawley rats differ in rapid cocaine-induced regulation of striatal dopamine transporter function. Neuropharmacology 58(3), 605–612.
- Mandt, B. H., Allen, R. M., & Zahniser, N. R. (2009). Individual differences in initial low-dose cocaine-induced locomotor activity and locomotor sensitization in adult outbred female Sprague-Dawley rats. Pharmacology, biochemistry, and behavior 91(4), 511–516.
- Gulley, J.M., Hoover, B.R., Larson, G.A., Zahniser, N.R. (2003). Individual differences in cocaine-induced locomotor activity in rats: behavioral characteristics, cocaine pharmacokinetics, and the dopamine transporter. Neuropsychopharmacology 28(12):2089-101.
- Laviola, G., Macrì, S., Morley-Fletcher, S., & Adriani, W. (2003). Risk-taking behavior in adolescent mice: psychobiological determinants and early epigenetic influence. Neuroscience and biobehavioral reviews 27(1-2), 19–31.
- Sengupta, P. (2013). The Laboratory Rat: Relating Its Age With Human's. Int J Prev Med 4(6):624-30.
- Spear, L.P. (2000). The adolescent brain and age-related behavioral manifestations. Neurosci Biobehav Rev 24:417–63.
